# Simplified molecular diagnosis of visceral leishmaniasis: Laboratory evaluation of miniature direct-on-blood PCR nucleic acid lateral flow immunoassay

**Norbert J. van Dijk**[1,2]*, **Dawit Gebreegziabiher Hagos**[1,2,3], **Daniela M. Huggins**[1], **Eugenia Carrillo**[4,5], **Sophia Ajala**[1], **Carmen Chicharro**[4,5], **David Kiptanui**[6], **Jose Carlos Solana**[4,5], **Edwin Abner**[6], **Dawit Wolday**[7], **Henk D. F. H. Schallig**[1,2]

1 Amsterdam University Medical Centre, Department of Medical Microbiology and Infection Prevention, Experimental Parasitology, Amsterdam, the Netherlands, 2 Amsterdam Institute for Infection and Immunity, Infectious Diseases Programme, Amsterdam, the Netherlands, 3 College of Health Sciences, School of Medicine, Department of Medical Microbiology and Immunology, Mekelle University, Mekelle, Ethiopia, 4 WHO Collaborating Centre for Leishmaniasis, National Center for Microbiology, Instituto de Salud Carlos III, Majadahonda (Madrid), Spain, 5 Centro de Investigación Biomédica en Red de Enfermedades Infecciosas (CIBERINFEC-ISCIII), Madrid, Spain, 6 Kacheliba Sub-County Hospital, Kacheliba, West Pokot County, Kenya, 7 Department of Biochemistry and Biomedical Sciences, McMaster University, Hamilton, Canada

* n.j.vandijk@amsterdamumc.nl

**Data Availability Statement:** Due to ethical regulations under which study participants were

## Abstract

### Background

Diagnosis of visceral leishmaniasis (VL) in resource-limited endemic regions is currently based on serological testing with rK39 immunochromatographic tests (ICTs). However, rK39 ICT frequently has suboptimal diagnostic accuracy. Furthermore, treatment monitoring and detection of VL relapses is reliant on insensitive and highly invasive tissue aspirate microscopy. Miniature direct-on-blood PCR nucleic acid lateral flow immunoassay (mini-dbPCR-NALFIA) is an innovative and user-friendly molecular tool which does not require DNA extraction and uses a lateral flow strip for result read-out. This assay could be an interesting candidate for more reliable VL diagnosis and safer test of cure at the point of care.

### Methodology/Principle findings

The performance of mini-dbPCR-NALFIA for diagnosis of VL in blood was assessed in a laboratory evaluation and compared with the accuracy of rK39 ICTs Kalazar Detect in Spain and IT LEISH in East Africa. Limit of detection of mini-dbPCR-NALFIA was 650 and 500 parasites per mL of blood for *Leishmania donovani* and *Leishmania infantum*, respectively. In 146 blood samples from VL-suspected patients from Spain, mini-dbPCR-NALFIA had a sensitivity of 95.8% and specificity 97.2%, while Kalazar Detect had a sensitivity of 71.2% and specificity of 94.5%, compared to a nested PCR reference. For a sample set from 58 VL patients, 10 malaria patients and 68 healthy controls from Ethiopia and Kenya, mini-dbPCR-NALFIA had a pooled sensitivity of 87.9% and pooled specificity of 100% using quantitative

enrolled and their samples collected and tested, the datasets used and/or analysed during the current study are only available upon reasonable request, directed to the Research Data Management unit of the Amsterdam UMC Research Support, Amsterdam, the Netherlands: rdm@amsterdamumc.nl. Dataset citation: van Dijk N, Hagos D, Schallig H. Diagnostic test results VL mini-dbPCR-NALFIA. DataverseNL; 2024.

**Funding:** This study was part of the EDCTP2 programme supported by the European Union: "Evaluation of the LAMP & db-PCR-NALFIA for the Diagnosis and/or as Test-of-Cure in Patients with Visceral Leishmaniasis in Ethiopia"; acronym "EvaLAMP & db-NALFIA"; grant number TMA2016SF- 1437; grant holder: DW; http://www.edctp.org/projects-2/. The funder had no role in study design, data collection and analysis, decision to publish, or preparation of the manuscript.

**Competing interests:** The authors have declared that no competing interests exist.

PCR as reference standard. IT LEISH sensitivity and specificity in the East African samples were 87.9% and 97.4%, respectively.

## Conclusions/Significance

Mini-dbPCR-NALFIA is a promising tool for simplified molecular diagnosis of VL and follow-up of treated patients in blood samples. Future studies should evaluate its use in endemic, resource-limited settings, where mini-dbPCR-NALFIA may provide an accurate and versatile alternative to rK39 ICTs and aspirate microscopy.

### Author summary

Visceral leishmaniasis (VL) is a severe infectious disease caused by unicellular parasites of the *Leishmania donovani* complex. As VL is fatal when left untreated, early and confirmatory laboratory diagnosis is crucial for adequate patient management. However, currently available serological assays, such as the rK39 rapid diagnostic test, fall short in terms of sensitivity and cannot be used for detection of relapsing infections. Therefore, we evaluated an innovative and user-friendly diagnostic assay called mini-dbPCR-NALFIA, which is based on the direct detection of *Leishmania* kinetoplast DNA in human blood samples. We found that mini-dbPCR-NALFIA can detect down to 500 parasites per millilitre of blood, which is lower than the parasite loads generally seen in untreated VL patients. Furthermore, we assessed the accuracy of mini-dbPCR-NALFIA using a set of Spanish and East-African samples from VL patients and control groups. In the Spanish samples, sensitivity of mini-dbPCR-NALFIA was very high and somewhat better than in the East-African samples. The specificity of mini-dbPCR-NALFIA was excellent in both sample sets. In conclusion, considering its robust performance and simplicity, the mini-dbPCR-NALFIA is a promising diagnostic assay that has the potential to improve VL diagnosis and follow-up of treated patients, especially in resource-limited situations.

## Introduction

Visceral leishmaniasis (VL), or "kala-azar", is a severe parasitic disease caused by an infection with the protozoan parasites *Leishmania donovani* or *Leishmania infantum*. These unicellular parasites are transmitted by blood-feeding female sandflies and infect phagocytes of the human host's reticuloendothelial system [1]. *L. donovani* is endemic in various regions in the Indian subcontinent and East Africa, while zoonotic transmission of *L. infantum* occurs in South America and Mediterranean countries. In clinical infections, VL presents as a serious illness with chronic fever, weight loss, hepatosplenomegaly and pancytopenia. When left untreated, the disease will be fatal in over 95% of the patients [2]. Accurate diagnosis is therefore key in the timely detection and management of VL.

Currently, easy-to-use and affordable rapid immunochromatographic tests (ICTs), detecting antibodies against *Leishmania* recombinant 39-amino acid repeat antigen (rK39) in capillary blood, are the first-line VL diagnostic in many endemic areas [3,4]. However, rK39 ICTs have a number of important shortcomings. For example, their performance has been shown to vary significantly between different geographical areas, and rK39 ICT sensitivity in East Africa is generally below 90% [3,5]. Furthermore, sensitivity of these serological tests is also reduced in immunocompromised VL patients, who often have impaired antibody production [6–8]. This is especially a problem in East Africa, where VL-HIV co-infection is common [9].

Thirdly, rK39 ICTs are unsuited for monitoring VL treatment effectiveness and detection of VL relapses, because *Leishmania* antibody levels usually remain elevated in blood long after initial infection and treatment [10,11]. As a consequence, the only available test of cure in many endemic countries remains microscopic examination of tissue aspirates for the presence of *Leishmania* amastigotes. However, this method is not preferable since it is only moderately sensitive and (incorrect) sampling of splenic aspirates may cause fatal haemorrhage [12,13].

Molecular techniques, such as polymerase chain reaction (PCR), have several advantages over microscopic and serological methods for VL diagnosis and follow-up. They generally have high sensitivity and specificity and can be applied on minimally invasive samples, like peripheral blood [14,15]. Moreover, molecular detection of parasite nucleic acids can be used to monitor therapeutic response and diagnose relapsing disease [16]. These assets have led to the implementation of PCR-based assays as reference test for VL diagnosis and confirmation of cure in many European countries. However, application of molecular techniques in endemic areas with limited resources is hampered by their complexity, costs and the requirement of advanced laboratory infrastructure, including sophisticated equipment for amplification and result read-out.

In recent years, several simplified molecular techniques have been developed to overcome these practical limitations of conventional PCR. One example is the loop-mediated isothermal amplification (LAMP), an isothermal DNA amplification assay for which the results can be read out with the naked eye thanks to fluorescence or turbidity caused by reaction by-products [17,18]. A recent meta-analysis showed that, compared to aspirate microscopy, LAMP has a sensitivity of 93.8% and specificity of 97.2% for detecting VL infections in blood samples [19]. Notwithstanding these promising results, a major drawback of LAMP remains the need of DNA extraction from the patient blood sample prior to testing [19]. Additionally, current LAMP protocols are not suited for multiplex target detection and therefore do not include amplification of a DNA extraction control. Implementation of LAMP has further been impeded by the high costs of the only commercially available *Leishmania* LAMP kit [20].

An alternative simplified molecular tool is the miniature direct-on-blood PCR nucleic acid lateral flow immunoassay (mini-dbPCR-NALFIA) [21,22]. Initially developed for diagnosis of malaria in resource-limited settings, this PCR-based method is applied directly on EDTA blood without the requirement of DNA extraction. The PCR is run on a miniPCR device, a portable miniature thermal cycler which is easily programmed through a smartphone application. By using 5'-labelled primers, the PCR products will carry a tag that will allow their detection with NALFIA, a dip strip with a fast and straight-forward read-out. A NALFIA strip has two antibody test lines, each directed against a unique amplicon tag, meaning that mini-dbPCR-NALFIA is suited for simultaneous detection of two different PCR targets. Its user-friendliness and minimal needs for laboratory equipment make mini-dbPCR-NALFIA a strong candidate for molecular diagnosis of VL in low-resource endemic settings. This study aimed to evaluate the performance of mini-dbPCR-NALFIA for detection of VL infection in blood samples from East Africa and Spain, using quantitative PCR (qPCR) and *Leishmania* nested PCR (LnPCR) as reference, respectively. Furthermore, mini-dbPCR-NALFIA accuracy was compared to that of the rK39 ICTs performed at the point of care: IT LEISH (Bio-Rad, Hercules, USA) in East Africa and Kalazar Detect (InBios, Seattle, USA) in Spain.

## Methods

### Ethical clearance

This study involved testing of clinical blood specimens from three different VL-endemic countries: Spain, Kenya and Ethiopia. For the use of clinical blood samples from the WHO

Collaborating Centre for Leishmaniasis (WHO-CCL) at the Carlos III Health Institute (ISCIII) in Madrid, Spain (registered at the National Biobank Register, Section Collections, Spain, reference ID: C.0000898), ethical clearance was obtained (ref. APR12–65 and APR14-64). For the collection and use of the blood samples from Kenya, ethical permission was obtained at the Amref Health Africa Ethics and Scientific Review Committee (ref. ESRC P1196/2022) as well as the National Association for Science, Technology and Innovation (ref. NACOSTI/P/22/18832) in Kenya. For the Ethiopian blood samples, the study received approval from the Institution Research Review Board of the College of Health Sciences, Mekelle University, Ethiopia (ref. ERC# -1102/2017). Both the Kenyan and Ethiopian review boards gave permission for the export of blood samples from the respective countries to the Laboratory for Experimental Parasitology at the Amsterdam University Medical Centre (AUMC, Amsterdam, The Netherlands), by means of a material transfer agreement that was signed by the providing and receiving institutions. Furthermore, samples from healthy Dutch blood donors that were deposited in a pre-established Biobank at the Laboratory for Experimental Parasitology at AUMC and collected in accordance with the Dutch Medical Research involving Human Subjects Act (ref. BWBR0009408), were used to determine the limit of detection of mini-dbPCR-NALFIA.

Before sample collection, written informed consent (and assent, in case of minors) for the collection and (secondary) use of samples were obtained from all participants and their respective parents or guardians. All samples used in this study were anonymized.

## Mini-dbPCR-NALFIA

Mini-dbPCR-NALFIA for diagnosis of VL in whole blood is based PCR amplification and subsequent lateral flow detection of a *Leishmania* kinetoplast DNA (kDNA) target, which was chosen because of its high copy number [18]. Additionally, the assay also amplifies and detects a human glyceraldehyde-3-phosphate dehydrogenase (GAPDH) gene target which serves as an internal amplification control. Briefly, 2.5 µL of EDTA anti-coagulated venous blood was lysed by heating at 98˚C for 10 minutes on a miniPCR mini16 (miniPCR bio, Massachusetts, USA). After lysis, 22.5 µL dbPCR reagent mix, consisting of 12.5 µL MyTaq Blood-PCR Mix (Meridian Bioscience, Cincinatti, USA), labelled forward and reverse primers for a kDNA target and GAPDH target (Eurogentec, Liège, Belgium) (Table 1) and nuclease-free sterile water, was added to the blood template. The used kDNA and GAPDH primers had been designed previously [23,24].

Next, direct on blood amplification was performed in a mini16 thermal cycler with the following protocol: 3 minutes at 95˚C; 30 cycles of 15 seconds at 95˚C, 30 seconds at 58˚C and 45 seconds at 72˚C; and a final 2 minutes at 72˚C. A NALFIA dipstick (Abingdon Health, York, UK) was used for read-out and placed in a mixture of 10 µL PCR product and 140 µL running buffer. During a 10-minute incubation, the labelled PCR amplicons flow over the NALFIA

**Table 1. Primer sequences, labels and final concentrations used in the direct-on-blood PCR.**

| Primer | Sequence | 5' labelling | Final concentration (nM) |
|---|---|---|---|
| kDNA Forward | 5'–TCCCAAACTTTTCTGGTCCT– 3' | Dig | 250 |
| kDNA Reverse | 5'–TTACACCAACCCCCAGTTTC– 3' | Biotin | 250 |
| Human GAPDH Forward | 5'–GAAGGTGAAGGTCGGAGTC– 3' | FAM | 150 |
| Human GAPDH Reverse | 5'–GAAGATGGTGATGGGATTTC– 3' | Biotin | 150 |

Dig: digoxigenin; FAM: fluorescein amidite.

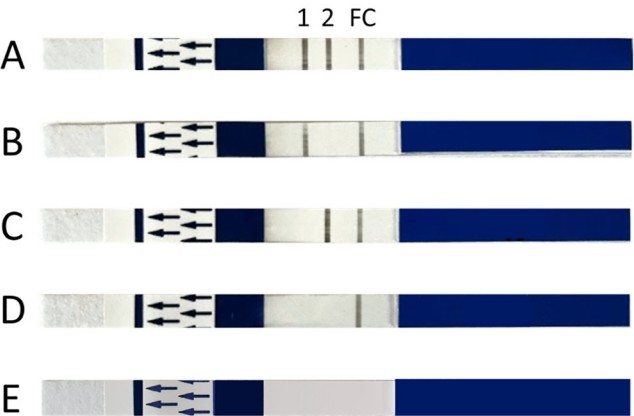

**Fig 1. Possible NALFIA results after dbPCR for *Leishmania* kDNA and human GAPDH.** The strip has three potential lines that can appear: a kDNA test line (1), a human GAPDH test line (2) and a flow control line (FC). (A) Both the kDNA and GAPDH lines are positive, indicating this is a VL-positive test result. (B) The GAPDH line is absent, however the kDNA line is positive, meaning that the PCR amplification was not inhibited. This result is considered valid and positive for VL. (C) A negative kDNA result and a positive GAPDH line, thus this test is negative for VL. (D) Both kDNA and GAPDH lines are absent, meaning that the dbPCR reaction was not successful. Therefore, this test result is invalid. (E) The NALFIA has no test lines nor a flow control line, meaning that this NALFIA test is invalid.

strip. They first pass the conjugate pad, where neutravidin-labelled carbon particles bind to the amplicons' biotin label. Next, the kDNA and GAPDH amplicons are captured by an anti-Dig and anti-FAM antibody line, respectively. Accumulation of captured amplicons resulted in the appearance of a visible test line (see Fig 1). A blood sample was considered to be positive for VL by the presence of a visible test line directed against Dig-labelled kDNA amplicons. A test was considered valid when the kDNA and/or GAPDH line were positive, together with a positive flow control line.

## Limit of detection

To determine the lowest *Leishmania* parasite density that is detectable with mini-dbPCR-NALFIA, serial dilutions of cultured *L. donovani* (strain WR352) and *L. infantum* (strain MHOM/TN/80/IPT-1) promastigotes in blood from a non-endemic individual were tested 20 times. *L. donovani* and *L. infantum* promastigotes were grown in RPMI 1640 medium with 10% foetal calf serum and 2% penicillin/streptomycin (both 10,000 units per mL) at 25˚C. Heat-inactivated culture isolates were used to prepare dilution series in EDTA blood from a healthy Dutch donor, with parasite densities ranging from $10^5$ to $10^1$ parasites per millilitre of blood. Ten-fold dilutions were tested first, followed by testing of intermediate parasite densities between the lowest positive and highest negative parasite densities to accurately determine the limit of detection (LoD). LoD was defined as the lowest parasite density that was detected with 95% confidence (i.e., at least 19 out of 20 tests).

## Sensitivity and specificity

The diagnostic accuracy of mini-dbPCR-NALFIA for detection of VL in EDTA whole blood was evaluated by applying this assay on two sample sets from different VL-endemic regions: Spain (endemic for *L. infantum*) and East Africa. The East African samples were collected in Ethiopia and Kenya (both endemic for *L. donovani*). Further details on the different sample sets are presented below.

**Sample size determination.** To calculate the required sample size for the mini-dbPCR-NALFIA evaluations in Spain and East Africa, the following assumptions were taken into consideration: taking 80% power, 5% marginal error, expected sensitivity and specificity 90%, and expected disease prevalence 50%, the minimum sample size was 134, 67 VL-positive patients and 67 VL-negative subjects, as determined by the reference standard [25].

**Sample sets.** For the evaluation of the performance of mini-dbPCR-NALFIA for detection of *L. infantum* infections, fully anonymized EDTA venous blood samples were retrieved from the Biobank of the WHO-CCL at ISCIII in Madrid, Spain. These samples had been collected from VL-suspected patients from *L. infantum*-endemic areas in Spain between 2019 to 2022. VL had been diagnosed by LnPCR targeting the *Leishmania* 18s rRNA gene, as described by Chicharro *et al.*, on DNA isolated from 200 µL of blood using the QIAamp DNA Mini Kit (QIAGEN, Hilden, Germany) [25]. This LnPCR is the diagnostic reference standard in WHO-CCL laboratory and has a LoD of 5 parasite equivalents (par-eq) per reaction [25]. Additionally, all samples had been serologically tested with Kalazar Detect (InBios, Seattle, USA) rK39 ICT. For the current study, 73 LnPCR-positive and 73 LnPCR-negative samples were randomly selected (see S1 Fig).

To evaluate mini-dbPCR-NALFIA for detection of *L. donovani* infections, a second set of samples comprised 136 EDTA venous blood samples from East-African subjects (see S2 Fig). Eighty-eight of these samples were selected from the Biobank of the Laboratory for Experimental Parasitology at AUMC, Amsterdam, the Netherlands, and had been collected from VL patients and healthy controls in northern Ethiopia between July 2019 to October 2020. The VL patients had been sampled at the Ayder Comprehensive Speciated Hospital in Mekelle and Kahsay Abera Hospital in Humera, Western Tigray, and had been diagnosed with VL based on clinical symptoms and a positive IT LEISH rK39 ICT (Bio-Rad, Hercules, USA). Healthy controls, free of VL-associated symptoms, came from endemic (Southern and Western Tigray, Zone 2 and 4 of the Afar region and Abergele-Sekota Woredas of Amhara region) and non-endemic regions (Mekelle city and neighbouring rural areas). At random, samples from 29 VL patients, 29 endemic controls and 30 non-endemic controls were selected. Of the selected controls, all had tested negative with IT LEISH ICT during sample collection except for 2 positive endemic subjects.

The remaining 48 East-African samples were prospectively collected for the purpose of this study in December 2022 in West Pokot County, Kenya, a region that is endemic for *L. donovani* [26]. Thirty-three samples were collected from febrile patients at the Kacheliba Sub-County Hospital: 22 samples from symptomatic VL patients with a positive IT LEISH rK39 ICT, 10 samples from IT LEISH-negative malaria patients (*Plasmodium falciparum* infection based on Giemsa-stained blood smear microscopy), and one sample from a patient that was positive for both IT LEISH ICT and *P. falciparum* blood slide. Furthermore, 15 samples were collected from healthy controls from local villages in West Pokot, which all tested negative with IT LEISH ICT at the time of sample collection. The 48 Kenyan samples were shipped to AUMC, Amsterdam, the Netherlands, for laboratory analysis.

All study subjects from Ethiopia were HIV-negative. The HIV status of study subjects from Spain and Kenya was unknown.

**Evaluation in Spanish samples.** The 146 Spanish samples were tested with mini-dbPCR-NALFIA at ISCIII, together with quantification of the parasite load of all LnPCR-positive samples with kDNA qPCR as follows: 1 µL of sample DNA eluate was added to 24 µL of qPCR reagent mix consisting of 12.50 µL 2x Rotor-Gene SYBR Green PCR Master Mix (QIAGEN, Hilden, Germany), kDNA forward and reverse primers (500 nM each) as described by Mary *et al.* [27], and DNase-free water. The qPCR was run on a Rotor-Gene Q (QIAGEN, Hilden, Germany) using the following protocol: 10 minutes at 95°C, followed by 40 cycles of 10

seconds at 95°C, 30 seconds at 60°C and 20 seconds at 72°C. A melting curve analysis was included to confirm specific target amplification. Each qPCR run included a standard curve of a 10-fold serial dilution of culture-extracted *L. infantum* genomes (JPCM5 strain, MCAN/ES/98/LLM-724) for quantification of the sample parasite densities. qPCR data analysis was performed using Rotor-Gene Q Software.

**Evaluation in East-African samples.** The Ethiopian and Kenyan sample sets were tested with mini-dbPCR-NALFIA at AUMC. An in-house qPCR, targeting *Leishmania* kDNA with a LoD of 0.025 par-eq per reaction and human GAPDH as DNA extraction control, was used as reference standard. For qPCR, DNA was isolated from 50 μL of the EDTA blood samples using automated extraction with the NucliSENS EMAG (bioMérieux, Marcy-l'Étoile, France). The isolated DNA was eluted in 25 μL EMAG elution buffer. Next, 1.25 μL of DNA eluate was tested with the PCR reagent master mix solution consisting of 6.25 μl of iTaq Universal Probes Master Mix (Bio-Rad, Hercules, USA), 3.85 μl of DNase-free water, 260 nM kDNA forward and reverse primers and 100 nM kDNA probe (developed by De Paiva-Cavalcanti *et al.* [23]), and 100 nM GAPDH forward and reverse primers and probe (developed by Hennig *et al.* [24]). The amplification was performed using Bio-Rad CFX96 real-time PCR which was programmed as follows: 10 minutes of UNG activation at 50°C, 5 minutes of initial denaturation at 95°C, and 40 cycles of 15 seconds of denaturation at 95°C and 45 seconds of annealing/elongation at 59°C. A Cq value < 36 was considered positive for kDNA and GAPDH based on in-house validation of the qPCR procedure. Each qPCR run included a standard curve of DNA extractions from a 10-fold serial dilution of *L. donovani* promastigote culture (strain WR352) diluted in blood, which was used to quantify the parasite densities in the tested samples. The qPCR data analysis was carried out using Bio-Rad CFX Maestro 2.0 software.

## Statistical analysis

All diagnostic test results were entered and analysed in Stata Statistical Software (Version 15, StataCorp LLC). The sensitivity and specificity of mini-dbPCR-NALFIA and rK39 ICT previously performed at the point of care were first calculated separately for the sample sets from the three different countries. For the samples from Spain, LnPCR was used as reference standard, since this is the VL diagnostic reference test at the WHO-CCL at ISCIII. kDNA qPCR was used as reference for the samples from Ethiopia and Kenya. Since these countries are both endemic for *L. donovani*, the pooled sensitivity and specificity of mini-dbPCR-NALFIA and IT LEISH ICT in East Africa were also calculated. The Clopper-Pearson method was applied for calculating the 95% confidence intervals (CI) of all sensitivity and specificity estimates. Cohen's κ was calculated to express the level of agreement beyond chance between mini-dbPCR-NALFIA, the reference tests and rK39 ICTs. The McNemar test was used to statistically compare the sensitivity of mini-dbPCR-NALFIA with the sensitivity of Kalazar Detect ICT in Spain and IT LEISH ICT in East Africa. The sensitivity of mini-dbPCR-NALFIA in Spain and East Africa, as well as the sensitivity of Kalazar Detect ICT in Spain and IT LEISH ICT in East Africa, were compared using Pearson's $\chi^2$ test. A P-value below 0.05 was considered statistically significant.

## Results

### Limit of detection

The LoD of mini-dbPCR-NALFIA for *L. donovani* was determined to be 650 promastigotes per mL of blood. For *L. infantum*, the detection limit was 500 promastigotes per mL. As the input for mini-dbPCR-NALFIA was 2.5 μL of blood, these LoD estimates corresponded to 1.625 and 1.25 par-eq/reaction for *L. donovani* and *L. infantum*, respectively.

**Table 2. Overview of the diagnostic test results for the samples included in this study.**

| | | Mini-dbPCR-NALFIA | | rK39 ICT[b] | |
|---|---|---|---|---|---|
| | | **Positive** | **Negative** | **Positive** | **Negative** |
| **Spain (N = 146[a])** | | | | | |
| LnPCR (reference standard) | Positive | 69 | 3 | 52 | 21 |
| | Negative | 2 | 70 | 4 | 69 |
| | Total | **71** | **73** | **56** | **90** |
| **Ethiopia (N = 88)** | | | | | |
| qPCR (reference standard) | Positive | 32 | 4 | 30 | 6 |
| | Negative | 0 | 52 | 1 | 51 |
| | Total | **32** | **56** | **31** | **57** |
| **Kenya (N = 48)** | | | | | |
| qPCR (reference standard) | Positive | 19 | 3 | 22 | 0 |
| | Negative | 0 | 26 | 1 | 25 |
| | Total | **19** | **29** | **23** | **25** |
| **East Africa (Ethiopia and Kenya pooled) (N = 136)** | | | | | |
| qPCR (reference standard) | Positive | 51 | 7 | 52 | 6 |
| | Negative | 0 | 78 | 2 | 76 |
| | Total | **51** | **85** | **54** | **82** |

[a] For mini-dbPCR-NALFIA, only the results for the 144 valid samples are displayed.

[b] Kalazar Detect for Spain, IT LEISH for Ethiopia and Kenya.

### Sensitivity and specificity

**Spanish sample set.** The diagnostic test results for all samples included in this study are summarized in Table 2, and depicted in S3 and S4 Figs. 73 LnPCR-positive EDTA blood samples from Spain had a median parasite density of $3.2 \times 10^3$ par-eq/mL of blood (interquartile range [IQR], $1.6 \times 10^3$–$1.1 \times 10^4$ par-eq/mL), as determined with qPCR. Of these, 69 had a positive kDNA test line result with mini-dbPCR-NALFIA (see Table 2 and S3 Fig), 3 were negative for kDNA with NALFIA and 1 was invalid due to a negative result for both the kDNA and GAPDH test lines. The NALFIA-negative samples had a parasite burden of 48, 113 and 991 par-eq/mL. Of the 73 samples with a negative LnPCR result, 70 were also negative for the kDNA NALFIA line, 2 samples showed a positive kDNA test line and 1 sample had an invalid NALFIA result.

Using the LnPCR as reference standard, and excluding the samples with an invalid NALFIA result from the analysis for this test, the sensitivity and specificity of mini-dbPCR-NALFIA were calculated to be 95.8% (95% CI, 88.3%–99.1%) and 97.2% (95% CI, 90.3%–99.7%), respectively, while the sensitivity and specificity of Kalazar Detect ICT were 71.2% (95% CI, 59.4%–81.2%) and 94.5% (95% CI, 86.6%–98.5%), respectively (Table 3). Mini-dbPCR-NALFIA in Spain was significantly more sensitive than Kalazar Detect ICT (P<0.001). The level of agreement between mini-dbPCR-NALFIA and LnPCR was almost perfect (Cohen's κ: 0.93), while the agreement between rK39 ICT and LnPCR was only substantial (Cohen's κ: 0.65). There was a considerable number of 21 samples that were negative for Kalazar Detect ICT but tested positive with LnPCR. Mini-dbPCR-NALFIA and Kalazar Detect ICT had a moderate agreement (Cohen's κ: 0.58).

**East-African sample set.** Among the 88 Ethiopian samples included in this study, all 29 IT LEISH ICT-positive VL patients were positive with both mini-dbPCR-NALFIA and qPCR testing at AUMC (see Table 2 and S4 Fig). Their median parasite concentration was $1.23 \times 10^4$

**Table 3. Accuracy estimates and Cohen's kappa for mini-dbPCR-NALFIA and rK39 ICT, compared to a PCR-based reference standard.**

| Sample set | Reference standard | Test | Sensitivity (95% CI) | Specificity (95% CI) | κ (95% CI) |
|---|---|---|---|---|---|
| Spain | LnPCR | Mini-dbPCR-NALFIA | 95.8% (88.3%– 99.1%) | 97.2% (90.3%–99.7%) | 0.93 (0.87–0.99) |
| | | Kalazar Detect | 71.2% (59.4%–81.2%) | 94.5% (86.6%–98.5%) | 0.65 (0.53–0.77) |
| Ethiopia | qPCR | Mini-dbPCR-NALFIA | 88.9% (73.9%–96.9%) | 100.0% (93.2%–100%) | 0.90 (0.70–1.00) |
| | | IT LEISH | 83.3% (67.2%–93.6%) | 98.1% (89.7%–100%) | 0.83 (0.62–0.99) |
| Kenya | | Mini-dbPCR-NALFIA | 86.4% (65.1%–97.1%) | 100.0% (86.8%–100%) | 0.87 (0.73–1.00) |
| | | IT LEISH | 100% (84.6%– 100%) | 96.2% (80.4%–99.9%) | 0.96 (0.88–1.00) |
| East Africa (Ethiopia + Kenya) | | Mini-dbPCR-NALFIA | 87.9% (76.7%–95.0%) | 100% (95.4%–100%) | 0.89 (0.82–0.97) |
| | | IT LEISH | 87.9% (78.8%–96.1%) | 97.4% (91.0%–99.7%) | 0.88 (0.80–0.96) |

par-eq/mL of blood (IQR, 6.4 x $10^2$–4.1 x $10^4$ par-eq/mL). From the healthy endemic control group, 7 out of 29 were positive for qPCR, of which 3 were also mini-dbPCR-NALFIA positive. The parasite load of the mini-dbPCR-NALFIA-negative but qPCR-positive endemic controls were 79, 202, 335 and 624 par-eq/mL of blood. On the other hand, all samples from the healthy volunteers in non-endemic areas were neither mini-dbPCR-NALFIA nor qPCR positive. Using qPCR as the reference standard, the sensitivity and specificity of mini-dbPCR-NALFIA in the Ethiopian samples were found to be 88.9% (95% CI, 82.3%–95.5%) and 100% (95% CI, 93.2%–100%), respectively. The level of agreement with the reference qPCR was almost perfect for mini-dbPCR-NALFIA (Cohen's κ: 0.90) and IT LEISH ICT (Cohen's κ: 0.83) (see Table 3), as was the agreement between mini-dbPCR-NALFIA and IT LEISH ICT (Cohen's κ: 0.88).

In the Kenyan sample set, 22 of the 23 samples from Kacheliba Hospital patients with clinical signs of VL and a positive IT LEISH ICT result were also kDNA positive with qPCR (see Table 2 and S4 Fig), with a median parasite load of 2.4 x $10^4$ par-eq/mL (IQR, 1.2 x 104–4.1 x $10^4$ par-eq/mL). Nineteen of the 22 qPCR-positive samples were positive with mini-dbPCR-NALFIA. This included a malaria patient with a qPCR-confirmed VL co-infection. The 3 qPCR-positive samples with a negative NALFIA result had parasite densities of 318, 795 and 883 par-eq/mL. The remaining 10 malaria patient samples and all 15 samples from the healthy endemic controls from West Pokot were negative with mini-dbPCR-NALFIA, IT LEISH ICT and qPCR. With the kDNA qPCR performed at AUMC as the reference standard, the sensitivity of mini-dbPCR-NALFIA was estimated at 86.4% (95% CI, 65.1%– 97.1%) while its specificity was 100% (95% CI, 86.7%– 100%). IT LEISH ICT sensitivity and specificity were 100% (95% CI, 84.6%– 100%) and 96.2% (95% CI, 80.4%–99.9%), respectively (see Table 3). Mini-dbPCR-NALFIA results were in almost perfect agreement with the qPCR results (Cohen's κ: 0.87) and IT LEISH ICT results (Cohen's κ: 0.83).

Combining the evaluation results for the Ethiopian and Kenyan samples, the pooled sensitivity and specificity of mini-dbPCR-NALFIA for VL diagnosis in East Africa were 87.9% (95% CI, 76.7%–95.0%) and 100% (95% CI, 95.4%–100%), respectively (see Table 3). The difference in mini-dbPCR-NALFIA sensitivity between East Africa and Spain (95.8%) was non-significant (P = 0.09). IT LEISH ICT had a pooled sensitivity of 87.9% (95% CI, 78.8%–96.1%) and a pooled specificity of 97.4% (95% CI, 91.0%–99.7%) in East Africa. The sensitivity of IT LEISH ICT in East Africa was significantly higher than the sensitivity of Kalazar Detect ICT (71.2%) in Spain (P = 0.009).

## Discussion

This study demonstrated that mini-dbPCR-NALFIA is a highly specific tool for diagnosis of VL and has excellent sensitivity for detection of *L. infantum* directly in blood. The sensitivity

of *L. donovani* was somewhat lower, hence further optimisation of the assay may be considered prior to its employment as test of cure for VL in East Africa and the Indian subcontinent.

Mini-dbPCR-NALFIA had a comparable LoD for *L. infantum* and *L. donovani*, namely 500 and 650 parasites/mL of blood, respectively. The small difference could be the result of different copy numbers of the kDNA target in the used strains [27,28]. The LoD of mini-dbPCR-NALFIA (1.625 to 1.25 par-eq/reaction) was relatively high compared to the reference kDNA qPCR at AUMC (0.025 par-eq/reaction), possibly due to lower efficiency of amplification directly on blood. For detection of minute parasite densities in treated patients, additional optimisation of mini-dbPCR-NALFIA to lower the detection limit should be considered, for example by adjusting the PCR reaction conditions and increasing the number of PCR cycles. Nevertheless, the majority of tested patient samples had a parasite density above the LoD, meaning that mini-dbPCR-NALFIA is potentially sensitive enough to be used for diagnosis of untreated VL in point-of-care settings with limited resources.

Mini-dbPCR-NALFIA had excellent specificity on both Spanish (97.2%) and East-African (100%) samples. Sensitivity was also high in Spain (95.8%), but appeared to be lower in East Africa (87.9%), although this difference was not statistically significant. Possible explanations include the genetic variation among the etiological agents and the highly sensitive qPCR reference standard used for the East-African samples. Moreover, the LoD of mini-dbPCR-NALFIA could be a limiting factor here, as East-African samples with false-negative NALFIA results often had blood parasite densities close to the established detection threshold for *L. donovani*.

In Spain, Kalazar Detect ICT had a poor sensitivity of 71.2%, which is in line with the previously reported sensitivity of 78.0% for this ICT brand in the same country [29]. IT LEISH ICT in East Africa performed better, with an overall sensitivity of 87.9%. As with the discrepant performance of mini-dbPCR-NALFIA in Spain and East Africa, differences in parasite species and patient population could play are role here. In addition, Kalazar Detect has often been shown to be less sensitive than IT LEISH [5,30–32]. The observed sensitivity of IT LEISH in East Africa was comparable to what has been reported before in Ethiopia (83.3%) and Kenya (89.3%) [5,33]. IT LEISH specificity of 97.4% was higher than in literature (82.2% in Ethiopia, 89.8% in Kenya), which may be due to differences in study population [5,33].

As mini-dbPCR-NALFIA was significantly more sensitive than Kalazar Detect ICT in Spain, this molecular diagnostic platform can be considered as a more reliable alternative for first-line VL diagnosis in this country, or as secondary test in case of a negative rK39 ICT result. Mini-dbPCR-NALFIA can be especially valuable at the primary health facilities such as small hospitals, health centres and health posts, where more complex PCR methods are unavailable [25]. Additionally, its use for simplified molecular diagnosis of canine leishmaniasis, a potential reservoir for zoonotic *L. infantum* transmission, could be a topic of further study [34]. The accuracy of mini-dbPCR-NALFIA in East Africa was similar to that of IT LEISH ICT, meaning that the former assay may need to be improved to compete with rK39 ICTs as primary VL diagnostic in this region. Nonetheless, mini-dbPCR-NALFIA remains a promising tool for VL diagnosis in HIV co-infected patients. Furthermore, in low-resource settings it may provide a highly needed alternative to invasive and unreliable aspirate microscopy for confirmation of cure after treatment and detection of relapsing disease.

The sensitivity of mini-dbPCR-NALFIA was comparable with previously reported sensitivity for LAMP (ranging from 83% to 98%) [19]. However, an important advantage of mini-dbPCR-NALFIA over LAMP, apart from the omission of DNA extraction, is its affordability: while LAMP costs per test can vary from 6 to 12 USD (depending on the used DNA extraction method), mini-dbPCR-NALFIA costs approximately 3 USD per test, which is comparable to rK39 ICTs [20,22,35]. The mini16 thermal cycler is also economical (800 USD per device),

meaning that mini-dbPCR-NALFIA is one of the first platforms that enables simplified molecular testing of VL in an affordable way.

The strength of this study lies in the evaluation of the mini-dbPCR-NALFIA on sample sets from different epidemiological settings, and the results demonstrate how the tool shows acceptable performance in both *L. infantum-* and *L. donovani-*endemic regions. The negative NALFIA results for the Kenyan samples from patients with malaria, another febrile protozoan infection which is often co-endemic with VL, further underline the specificity of the assay. Moreover, mini-dbPCR-NALFIA performed well at both the laboratories of AUMC and ISCIII: of the 282 samples tested in this study, only 2 (0.7%) had an invalid result, possibly due to an error in PCR preparation for these samples.

While the number of samples from East Africa was sufficient to make a precise estimation of mini-dbPCR-NALFIA performance in this region, a sensitive per-country analysis was hampered by the relatively small sample sizes for the individual countries. Another limitation of this study was the use of different reference standards by the evaluations at ISCIII and AUMC, although both LnPCR and qPCR have been shown to be highly accurate methods for diagnosing VL [18,25]. The kDNA qPCR at AUMC did not include a melting curve analysis, meaning that incidental false-positive signals due to non-specific products cannot be excluded.

In conclusion, mini-dbPCR-NALFIA is a simplified molecular diagnostic test for VL, with excellent performance for detecting *L. infantum* infections in patients from Spain, where it provides an affordable and more sensitive alternative to the underperforming rK39 ICT. In East Africa, the acceptable sensitivity of mini-dbPCR-NALFIA make this tool an interesting candidate to replace aspirate microscopy as test of cure and VL relapse diagnostic. These applications should be evaluated in future studies, which may also attempt to lower the detection limit and thereby increase the sensitivity, in particular for *L. donovani*.

## Supporting information

**S1 Fig. Selection of Spanish study samples.**
(TIF)

**S2 Fig. Selection of East-African study samples.**
(TIF)

**S3 Fig. Venn diagram of the positive diagnostic test results for the Spanish study samples.**
(TIF)

**S4 Fig. Venn diagram of the positive diagnostic test results for the East-African study samples.** ET: Ethiopia; KE: Kenya.
(TIF)

## Acknowledgments

We would like to express our acknowledgements to Mekelle University, College of Health Sciences and University of Amsterdam, Academic Medical Centre for their technical support during sample collection and laboratory analysis. We would like to thank the WHO Collaborating Center for Leishmaniasis at ISCIII in Madrid, Spain, led by Dr. Javier Moreno, for collaborating in this research and the support during the development and validation of mini-dbPCR-NALFIA for VL. Our gratitude also goes to the clinical staff at the Kacheliba Sub-County Hospital in Kenya and at the Ayder Comprehensive Speciated Hospital, Mekelle and Kahsay Abera Hospital, Humera, in Ethiopia for the collection of the East-African patient and

control samples. Last but not least, we want to thank all the study participants for their voluntary participation in the study.

## Consent to publish

All human subjects from which the samples used in this study were derived gave their written informed consent to publish the outcomes of the analysis on their samples.

## Author Contributions

**Conceptualization:** Norbert J. van Dijk, Daniela M. Huggins, Eugenia Carrillo, Henk D. F. H. Schallig.

**Data curation:** Norbert J. van Dijk, Dawit Gebreegziabiher Hagos, Daniela M. Huggins.

**Formal analysis:** Norbert J. van Dijk, Dawit Gebreegziabiher Hagos, Daniela M. Huggins, Jose Carlos Solana.

**Funding acquisition:** Dawit Wolday.

**Investigation:** Norbert J. van Dijk, Dawit Gebreegziabiher Hagos, Daniela M. Huggins, Sophia Ajala, Carmen Chicharro, David Kiptanui, Edwin Abner.

**Methodology:** Norbert J. van Dijk, Dawit Gebreegziabiher Hagos, Daniela M. Huggins, Carmen Chicharro, Jose Carlos Solana.

**Resources:** Eugenia Carrillo, Dawit Wolday, Henk D. F. H. Schallig.

**Supervision:** Norbert J. van Dijk, Eugenia Carrillo, Dawit Wolday, Henk D. F. H. Schallig.

**Writing – original draft:** Norbert J. van Dijk, Dawit Gebreegziabiher Hagos.

**Writing – review & editing:** Eugenia Carrillo, Carmen Chicharro, Henk D. F. H. Schallig.

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
