## [Decision Letter · Decision Letter 0]

10 Jan 2024

Dear Mr. van Dijk,

Thank you very much for submitting your manuscript "Simplified molecular diagnosis of visceral leishmaniasis: development and laboratory evaluation of mini-dbPCR-NALFIA" for consideration at PLOS Neglected Tropical Diseases. As with all papers reviewed by the journal, your manuscript was reviewed by members of the editorial board and by several independent reviewers. In light of the reviews (below this email), we would like to invite the resubmission of a significantly-revised version that takes into account the reviewers' comments. 

We cannot make any decision about publication until we have seen the revised manuscript and your response to the reviewers' comments. Your revised manuscript is also likely to be sent to reviewers for further evaluation.

Sincerely,

Daniel K. Masiga

Academic Editor

Abhay Satoskar

Section Editor

Reviewer's Responses to Questions

**Key Review Criteria Required for Acceptance?**

**Methods**

-Are the objectives of the study clearly articulated with a clear testable hypothesis stated?

-Is the study design appropriate to address the stated objectives?

-Is the population clearly described and appropriate for the hypothesis being tested?

-Is the sample size sufficient to ensure adequate power to address the hypothesis being tested?

-Were correct statistical analysis used to support conclusions?

-Are there concerns about ethical or regulatory requirements being met?

Reviewer #1: Yes,

Though I have a query regarding 5' mismatches in the primers used, which may explain differences in sensitivity in Spain vs. East Africa. The query is within the manuscript.

Reviewer #2: ABSTRACT

- In the overall rationale and aim of the study, I think it is important to highlight the invasiveness of the tissue aspirate microscopy, for which the dbPCR-NALFIA could be a solution, rather than focusing on the POC test. 

- Some sections of the abstract can be shortened and instead the diagnostic accuracy of the rK39 RDTs (brand specified) can be added. This especially when authors would still conclude from their study that their test can replace the rK39 RDT. 

AUTHOR SUMMARY

- Line 51: Also in the Americas. 

- Line 52: What do the authors mean by “optimum treatment outcomes”, a good prognosis? 

- Lines 57-59: I do not really think that 500 par/mL is a very low parasitemia, especially for TOC/treatment follow-up you would want a test that can see whether all parasites are cleared from the blood or not. 

INTRODUCTION

General remarks:

- The background section is rather long and contains information that is not necessarily important to understand the content of the manuscript. For example, the authors explain in depth the impact of the disease in East-Africa, then again in the Mediterranean and then focus on outbreaks in southern Europe (nothing about Indian subcontinent?). Please try to reduce this part. Most important is that there is two main causative species in different parts of the world and that due to the elimination initiative in the Indian subcontinent majority of cases are now reported from East-Africa, where available diagnostics have major limitations. 

- Authors mention in Line 84 that the gold standard for VL diagnosis is demonstration of Leishmania parasites in tissue aspirates through microscopy, however, this is only the case in resource-limited countries. I also do not completely agree that this is the gold standard, as it is only the last tool that will be employed if the other available methods in the diagnostic algorithm fail to detect Leishmania infection. Please revise. 

- Related to the comments above, the section about tissue aspiration is too lengthy and needs to be reduced. In line 89, The authors state that tissue microscopy is unsuited for rural settings, however, in general because of the invasiveness and low sensitivity of the method it is not a preferable approach. 

- Overall I think the background would benefit from a shift in the structure, following the diagnostic algorithm. 

- An important advantage of the use of molecular tools is that it can be employed on less-invasive samples like blood. 

- The authors mention that molecular methods are not employed yet with the focus on East-Africa, however, to my knowledge it is also not routinely employed in the Americas or Indian subcontinent. The authors should either broaden the scope or highlight that the rK39 RDT diagnostic accuracy is suboptimal in East-Africa and therefore diagnosis often relies on tissue aspiration which does not allow for decentralization of diagnosis and care. 

- The authors could still highlight that in Sudan, it was recommended to incorporate LAMP in the diagnostic algorithm for VL, before tissue microscopy would be done. However, major limitations to do so are indeed the mentioned disadvantages, but I think also still the cost of a LAMP assay. 

- In line 126, can the authors specify that one target is to check for DNA extraction efficiency and indicate which Leishmania marker is targeted? kDNA? If so, maybe highlight why kDNA was selected as target. 

- Perhaps good to briefly highlight the diagnostic accuracy found so far of the mini-dbPCR-NALFIA, and then proceed with the rationale of the study, which is not clearly described now. Furthermore, from the overall objective it is currently not clear what the reference test was. A composite reference of rK39 + qPCR or nPCR? This is not clear, please clarify. 

METHODS

Ethics

Samples have been exported out of East-Africa for analysis in the Netherlands. Can the authors specify whether (national?) IRBs approved this and indicate whether there was an MTA allowing this? 

General comments

- More information needs to be provided on the source of the samples/study population. Was it secondary use of biobanked samples or fresh samples? If the former, did patients consent to the use of their samples for secondary research? What did the patient population look like? What was used as a definition of VL? Was any sample size calculation performed? Please elaborate futher on study population and sample collection/storage in a separate paragraph. 

- It seems like part of the methods section is missing after line 162. What were the reaction conditions used? 

- Can the authors explain why they did not go lower than 10 par/mL for the LoD? The LoD of kDNA is known to be much lower than that. 

- Why are the results of the rK39 RDT already mentioned in the methods if this was a study procedure and so a result? 

- Perhaps it would help to have a flow chart or table in the methods to clarify the sample sets used for testing: where samples came from (especially in East-Africa: it is strange it comes up in the last sentence only and is still unclear), what was considered a VL-suspected patient, whether it was a prospective collection on secondary use of samples (in case of the latter, how were samples selected considering that authors knew which ones were – strongly – positive and selected exactly half negatives/positives), how they were tested previously/for the study etc. Going through the text, it is not clearly described and numbers do not seem to add up but that is perhaps due to the fact that the way it is described is a bit confusing. It would be useful if the authors considerably revise this section and add a figure or table to guide it. 

- Can the authors clarify the cut-off of 36 they used for positivity of the kDNA PCR or refer to a paper that describes this?

**Results**

-Does the analysis presented match the analysis plan?

-Are the results clearly and completely presented?

-Are the figures (Tables, Images) of sufficient quality for clarity?

Reviewer #1: Yes

Reviewer #2: RESULTS

- It is not entirely clear to me how the authors calculated with this much depth the LoD of the mini-dbPCR-NALFIA that the difference between 500 and 650 par/mL can be detected. 

- Line 385: were these samples that were mini-dbPCR-NALFIA positive also positive by kDNA PCR? 

- To show the agreement between the different tests, instead of only using values in the text, authors could perhaps add a (supplementary) VENN diagram to show the overlapping results in the three sample sets. 

FIGURES AND TABLES

- The authors should ensure that the figures and tables are presented in the text at the right time and should link to it in their text. 

- The authors indicate in figure 1 that a band for kDNA but not for GAPDH is still considered positive. Is that a likely possibility? Could it be contamination is this case?

- There is no reference in the text to Table 2

**Conclusions**

-Are the conclusions supported by the data presented?

-Are the limitations of analysis clearly described?

-Do the authors discuss how these data can be helpful to advance our understanding of the topic under study?

-Is public health relevance addressed?

Reviewer #1: Yes

Reviewer #2: - As throughout the rest of the text, also here the authors can be more comprehensive. The first Alinea of the discussion is repetitive from the introduction and repeats again what the overall aim of the study was. Methods used are explained again and results are repeated. Authors should focus more on how results should be interpreted and placed in a broader context. 

- Authors mention that further optimization could be done to improve the LoD but do not mention what can be done to indicate that indeed there is potential that it could be improved. I was a bit surprised to see that kDNA, which normally has a very low LoD in PCR, resulted in a high LoD. Is this most likely due to the direct PCR on blood without extraction or due to the read-out with the lateral flow? Can the authors reflect on this? Also the word “could” should probably be “should”, because with the current diagnostic accuracy I do not see how otherwise it would ever be implemented in routine practice in the field. 

- The authors state: “The lower sensitivity in East Africa may have resulted from parasite densities close to the LoD of mini-dbPCR-NALFIA, found in a number of samples and leading to a false-negative NALFIA result.” These data are available from the qPCR right? Can the authors add these data to the results and make the statement stronger? 

- It is mentioned that the InBios RDT in Spain had a considerably lower sensitivity than the dbPCR-NALFIA, which was not the case in East-Africa (where IT-Leish is used). Could this be intrinsic to the diagnostic accuracy of the specific RDT brand? Can the authors put this finding in a broader context and compare the obtained sens/spec with the literature. Or is it possible that if the RDT was used on stored samples that its sensitivity decreased as it’s normally employed as a POC test? To clarify this, authors should refer more to existing literature. 

- In the section about the employment of the dbPCR- NALFIA in East-Africa the authors should focus their statement particularly on the test-of-cure and diagnosis of relapsing patients. Except if the sensitivity can greatly be increased, it will probably not be employed in EA. However, for example in northwest Ethiopia, there is a very high proportion of HIV-VL coinfected patients, who still frequently undergo an invesive TOC and are repeatedly relapsing. For such populations, its employment would mostly be beneficial. 

- It would be good if the authors can explain in the manuscript what the cost is of the current test, including instruments needed. In order for the test to be cost-beneficial, also in Spain, it should not only be easy and have good diagnostic accuracy, but also be cheap, unless of course if it would only be used for few patients. This is an important operational challenge that is often not described, but critical information for decision makers. 

- The strengths and weaknesses of the study can be written more concise. 

- In the conclusion, rather than using the test for POC diagnosis in place of the rK39 RDT, I think more focus can be put on its use for test-of-cure and relapsing patients, and so to get rid of the highly invasive aspirates for patients. Furthermore, authors can be more careful with their statement about Spain as well if the costs are higher than the lateral flow assays, because then take-up of the dbPCR-NALFIA would probably still not be as a first-line diagnosis, but maybe when the rK39 RDT is negative and there is still high suspicion of VL. 

- Another reflection, however, is that as a test-of-cure one would like to have a very sensitive test, because even blood parasite load does not mean that the patient is completely cleared. So even though the dbPCR-NALFIA is less invasive, its LoD is still not really optimal for TOC and so should be further improved indeed. 

- The authors describe a couple of times in their text that the test under validation is very robust. However, it did not provide valid results for some of the samples, which is not discussed anymore in the rest of the manuscript, but which indicates that it is not completely robust. Can the authors further elaborate on this and maybe compare this with the validity of other tests employed in routine?

**Editorial and Data Presentation Modifications?**

Reviewer #1: I have made minor edits and suggestions in the attached track-changes document.

Reviewer #2: BACKGROUND

- Line 68: Check sentence construction “caused by infection with protozoan Leishmania parasites”

- Line 85: Please change “aspirates of infected tissues” to “tissue aspirates”

- Line 93-96: Try to reduce the length of the sentence, e.g. “They are unsuitable for monitoring of VL treatment response and diagnosis of VL relapse”.

- Line 98: “Where VL-HIV co-infection is common”. Also no need to mention again that the rK39 sensitivity is significantly lower, that is already mentioned in the previous sentence. 

- Line 121: I would recommend to change DNA purification to DNA isolation or extraction

- Line 128: I would advice the authors not to focus too much on the “rural” endemic settings. Maybe they are referring to low-resource, endemic settings? I assume that the test is not appropriate yet to bring to the field during outreach campaigns as is done with the rK39 ICTs.

METHODS

- Lines 154-155: I feel this information should be in the background section. The part 

---

## [Editor Report · Decision Letter 1]

15 Apr 2024

Dear Mr. van Dijk,

We are pleased to inform you that your manuscript 'Simplified molecular diagnosis of visceral leishmaniasis: laboratory evaluation of miniature direct-on-blood PCR nucleic acid lateral flow immunoassay' has been provisionally accepted for publication in PLOS Neglected Tropical Diseases.

Best regards,

Paul Brindley

Editor in Chief

---

## [Editor Report · Acceptance letter]

19 Apr 2024

Dear Mr. van Dijk,

We are delighted to inform you that your manuscript, "Simplified molecular diagnosis of visceral leishmaniasis: laboratory evaluation of miniature direct-on-blood PCR nucleic acid lateral flow immunoassay," has been formally accepted for publication in PLOS Neglected Tropical Diseases.

Best regards,

Shaden Kamhawi

co-Editor-in-Chief

Paul Brindley

co-Editor-in-Chief
